# Simultaneous Quantitative Determination of Low-Concentration Preservatives and Heavy Metals in Tricholoma Matsutakes Based on SERS and FLU Spectral Data Fusion

**DOI:** 10.3390/foods12234267

**Published:** 2023-11-26

**Authors:** Yuanyin Jin, Chun Li, Zhengwei Huang, Ling Jiang

**Affiliations:** College of Information Science and Technology, Nanjing Forestry University, 159 Longpan Road, Nanjing 210037, China; 15720612874@163.com (Y.J.); chunli0205@njfu.edu.cn (C.L.); nupt_hzw@hotmail.com (Z.H.)

**Keywords:** fluorescence spectroscopy, surface-enhanced Raman spectroscopy, data fusion, potassium sorbate, lead element

## Abstract

As an ingredient of great economic value, Tricholoma matsutake has received widespread attention. However, heavy metal residues and preservatives in it will affect the quality of Tricholoma matsutake and endanger the health of consumers. Here, we present a method for the simultaneous detection of low concentrations of potassium sorbate and lead in Tricholoma matsutakes based on surface-enhanced Raman spectroscopy (SERS) and fluorescence (FLU) spectroscopy to test the safety of consumption. Data fusion strategies combined with multiple machine learning methods, including partial least-squares regression (PLSR), deep forest (DF) and convolutional neural networks (CNN) are used for model training. The results show that combined with reasonable band selection, the CNN prediction model based on decision-level fusion achieves the best performance, the correlation coefficients (*R*^2^) were increased to 0.9963 and 0.9934, and the root mean square errors (*RMSE*) were reduced to 0.0712 g·kg^−1^ and 0.0795 mg·kg^−1^, respectively. The method proposed in this paper accurately predicts preservatives and heavy metals remaining in Tricholoma matsutake and provides a reference for other food safety testing.

## 1. Introduction

As a nutritious and precious ingredient, Tricholoma matsutakes have antioxidant, immune-boosting, anti-inflammatory and blood sugar-regulating properties [1]. However, they are highly susceptible to pollutants in the environmental soil during their growth, such as burdensome metal elements [2,3]. The root and filamentous mycelium system of Tricholoma matsutakes can absorb lead, cadmium, and mercury in the soil. Apart from heavy metal contamination, the quality of Tricholoma matsutakes may also be affected by excessive preservatives added by traders during transportation to preserve freshness. Long-term consumption of Tricholoma matsutake with excessive heavy metals and preservatives will affect the digestive system of the human body, and heavy metal poisoning may also occur, posing significant challenges to food safety [4]. The accumulation of preservatives and heavy metals in the body may lead to an acid-base imbalance in the human body, causing symptoms such as dizziness and diarrhea. In severe cases, it may cause chronic poisoning and increase the risk of cancer. To maintain good health, people should pay attention to controlling their daily intake of preservatives and heavy metals. Therefore, the detection of preservatives and heavy metal content plays a vital role in the quality control of Tricholoma matsutakes and the guarantee of food safety. With the increased awareness of the health concept, accurate and efficient detection of Tricholoma matsutake quality and quantitative analysis of illegally used additives have become hot topics in modern medicine and food. In recent years, conventional methods based on graphite furnace atomic absorption spectrometry and liquid/gas chromatography have been widely used in the analysis of preservatives and heavy metals in Tricholoma matsutakes [5,6]. Although the accuracy of these traditional methods is relatively high, the corresponding time and labor costs cannot be ignored, which is far from meeting the needs of rapid detection [7]. Traditional methods are often limited to high-precision detection of a single component, which cannot simultaneously detect the content of preservatives and heavy metals in Tricholoma matsutakes.

Recently, spectroscopy technology has gradually emerged in the technical field of preservatives and heavy metal detection with its advantages of fast detection and low sample loss. Yang, et al. established a potassium sorbate content in cocktails predictive model based on surface-enhanced Raman spectroscopy (SERS) [8]. The root mean square error (*RMSE*) of the model is 0.1429 g·kg^−1^, and the limit of detection (*LOD*) can reach 0.062 g·kg^−1^. Wang, et al. used an improved chicken swarm optimization support vector machine (ICSO-SVM) combined with three-dimensional fluorescence (FLU) spectra to rapidly detect the potassium sorbate ranges 0.007 to 0.1 g·L^−1^ in orange juice [9]. The best model result of mean square error (MSE) is 1.01·10^−5^ g·L^−1^. Spectroscopic technology has been verified to have great application prospects in the qualitative analysis of potassium sorbate. However, these existing studies still focus on detecting single-added substances, while additives do not exist alone in reality. SERS-based methods require considerable efforts to develop corresponding substrates for analytes to enhance the Raman signal and improve the accuracy of prediction. Compared with the SERS-based method, the FLU-based method can achieve higher sensitivity and resolution during measurement. Affected by factors such as scattering, self-absorption and temperature, the instability of fluorescence methods at high concentrations will limit the accuracy of the prediction model. With the increased emphasis on food safety and the strict restrictions on the content of food additives, achieving simultaneous detection of multiple mixed substances while ensuring detection accuracy has become a hot spot in the current field of food testing. Therefore, spectral data fusion technology is used in this work to make up for the shortcomings of single spectral methods to establish a simultaneous detection model for preservatives and heavy metals.

As a framework for integrating multi-source sample input signals, spectral data fusion takes advantage of the complementary synergistic advantages of different input information to significantly make up for the shortcomings of a single spectral data source. Data fusion techniques have been widely used in the quantitative analysis of multiple indicators [10,11,12]. Zhao, et al. used near-infrared (NIR) and laser-induced breakdown spectroscopy (LIBS) to quantitatively analyze the heavy metals in lily [13]. The introduction of near-infrared spectroscopy makes up for the inability of LIBS to accurately quantify complex matrix samples. Compared with the full-spectrum model, the model based on feature-level fusion achieves better performance in quantifying Zn, Cu and Pb, with *R*^2^ of 0.9858, 0.9811 and 0.9460, and *RMSE* of 4.3047 mg·kg^−1^, 4.9592 mg·kg^−1^ and 0.9460 mg·kg^−1^. Li, et al. used visible near-infrared (Vis-NIR) and near-infrared (NIR) to qualitatively assess total volatile basic nitrogen (TVB-N) and total viable count (TVC) in chickens [14]. With the introduced data-level and feature-level fusion strategy, the root mean square error of prediction (*RMSEP*) in TVC and TVB-N content can reach 0.1889 and 2.6094, respectively. Compared with the anticipated results based on single spectra, the *RMSEP* values decreased by 0.0087 and 0.2816, respectively. Yang, et al. performed a quantitative analysis of adulterated honey by combining spectral analysis with multiple high-level data fusion strategies [15]. Three decision-level fusion strategies based on binary linear regression, entropy weight method and trend line slope weight method were adopted, which achieved better results compared with full-spectrum and feature-level fusion strategies. Based on the fusion of UV–Vis and NIR spectral data, Xu, et al. proposed an alternative approach for simultaneous detection of chemical oxygen demand (COD), ammonia nitrogen (AN) and total nitrogen (TN) detection in surface water [16]. With the introduced data fusion strategy, the *RMSEP* of the three parameters can reach 6.95, 0.195 and 0.466, respectively, which is decreased by 2.96%, 11.3% and 4.23% compared with single-spectroscopic-based models. The studies mentioned above have proved that appropriate data fusion strategies can effectively improve the results in the quantitative detection of multivariate mixtures.

In this work, a method based on SERS and FLU spectroscopy technology for the simultaneous determination of potassium sorbate and the main heavy metal element lead in Tricholoma matsutake was proposed to replace the traditional detection methods. Through appropriate waveband selection and sample preprocessing, multivariate mixed detection is converted into a quantitative analysis of two single substances to improve prediction accuracy. Moreover, the complementarity of SERS and FLU spectral detection methods is exploited to further optimize the quantitative detection model through a decision-level fusion strategy. Existing research on preservatives and heavy metal spectral detection mainly focuses on single-spectrum analysis and model optimization. With our method, the amount of spectral data and prediction accuracy are significantly optimized, and the *LOD* is minimized. The results of this study try to provide a theoretical basis for high-end food quality assessment.

## 2. Materials and Methods

### 2.1. Sample Preparation

The Tricholoma matsutakes were collected from Kunming, Yunnan province, and selected with different sizes and shapes. Collected Tricholoma matsutakes were cleaned with ultrapure water and homogenized into small particles by a ceramic knife. The potassium sorbate aqueous solution and lead standard solution were added proportionally to the cleaned and homogenized samples to simulate contamination of preservatives and heavy metal elements. We added acetonitrile and extraction salt to the homogenized samples, took the supernatant as the Tricholoma matsutake extract, and vortexed the extract in the purification tube to eliminate fluorescence interference. Because the purchased Tricholoma matsutake samples originally contained lead elements, the extract with low lead concentrations cannot be obtained directly. A selected portion of extract samples was subjected to be extracted and purified, and lead ions were adsorbed from the extracting solution using a composite material. The processed extracting solution without lead was used to dilute other extracts to obtain the desired low concentrations. The standard for potassium sorbate content in mushrooms in China is no more than 0.5 g·kg^−1^, and the standard for lead content is no more than 1 mg·kg^−1^. Samples were prepared into 15 potassium sorbate concentrations (0 to 2 g·kg^−1^) and 15 lead concentrations (0 to 2 mg·kg^−1^). The selected sample concentrations are shown in Table 1. By combining different concentrations of potassium sorbate and lead, 225 samples with different potassium sorbate and lead contents were prepared in the experiment. Ten samples of each type were prepared to ensure the credibility of the experimental results.

### 2.2. Spectroscopy Data Acquisition

Raman spectroscopy is a powerful label-free technique to identify molecules by measuring the vibrational and rotational character of their chemical bonds. SERS exploits the phenomenon of enhanced Raman scattering on the surface of plasmonic nanoparticles or nanostructures [17]. The SERS spectra of Tricholoma matsutakes samples were acquired by a laser SERS spectrometer (DXR532) with a 785 nm laser source equipped with a coupled device detector. During the SERS experiments, modified gold nanoparticle sol was used as the substrate to simultaneously enhance the SERS signals of potassium sorbate and lead ions. The excitation wavelength of the light source was set to 780 nm, with a power of 150 mW and a resolution of 4. The spectral scanning range was set from 50 to 3000 cm^−1^. To ensure the accuracy of the experiment, three spectra were collected for each sample, and the average spectra were obtained as the final result.

Fluorescence is a type of radiation transition, which is the radiation released by a substance from an excited state to a low-energy state with the same multiplicity. When a molecule in the ground state absorbs energy and jumps, the molecule becomes unstable and jumps back to the ground state. Photons are emitted during the transition back to the ground state, which produces fluorescence [18]. The FLU spectra were measured by a steady-state/lifetime spectrofluorometer. The slits of excitation/emission were set at 3 nm. To achieve a quantitative analysis of lead elements, the excitation wavelength was set as 262 nm, and the emission wavelength range was 300–500 nm. When the excitation wavelength is set to 262 nm, the emission spectra change intensity is not affected by changes in potassium sorbate content. To achieve a quantitative analysis of potassium sorbate, the excitation wavelength was set as 358 nm, and the emission wavelength range was 375–700 nm. Considering that the influence of lead on the emission spectra of FLU cannot be eliminated even by changing excitation wavelength, the extract was purified by composite materials before testing to ensure accurate measurement of potassium sorbate. All the samples were scanned three times to reduce instrumental errors, and the average spectra were obtained as the final result.

### 2.3. Data Analysis Methods

#### 2.3.1. Quantitative Models and Evaluation

To quantitatively analyze the lead element and potassium sorbate in Tricholoma matsutakes, three algorithms of partial least-squares regression (PLSR), deep forest (DF) and convolutional neural networks (CNNs) were used to establish the regression model [19]. PLSR is a regression modeling method from multiple dependent variables to multiple independent variables. Deep forest, as an ensemble method of decision trees, exhibits strong competitiveness compared to deep neural networks (DNN) and is much easier to train [20]. Convolutional neural networks (CNNs) are a class of feedforward neural networks that incorporate convolutional computations with a deep structure.

To better evaluate the prediction performance, the determination coefficient (*R*^2^) between the reference and predicted value, *RMSE* and mean absolute error (*MAE*) were applied here:(1)$$R^{2}=1-\sum_{i=1}^{n}\;{\left(y_{i}-y_{\mathrm{predi}}\right)}^{2}/\sum_{i=1}^{n}\;{\left(y_{i}-y_{\mathrm{mean}}\right)}^{2}$$
(2)$$RMSE=\sqrt{\sum_{i=1}^{n}\;{\left(y_{\mathrm{predi}\;}-y_{i}\right)}^{2}/n}$$
(3)$$MAE=1/n\times \sum_{i=1}^{n}\;\left|y_{\mathrm{predi}\;}-y_{i}\right|$$
where *n* is the number of fitting points, $y_{i}$, $y_{\mathrm{mean}}$ and $y_{\mathrm{predi}\;}$ are the actual value, average value and the predicted value of the concentration, respectively.

#### 2.3.2. Data Processing and Feature Extraction

The original spectral data are usually unsuitable for direct modeling analysis due to noise, background interference and experimental operating errors. In this study, the collected SERS spectral data are standardized to correct errors caused by variations in the focal distance during the data acquisition.

To increase the amount of useful information on the spectra, and improve the resolution and signal-to-noise ratio of the spectra, Gramian angular field (GAF) [21], Markov transition field (MTF) [22], relative position matrix (RPM) [23] and recurrence plots (RP) transformation methods were used to transform the SERS and FLU spectra into 2D spectrograms [24]. The obtained 2D spectral data were subsequently used to develop 2D CNN regression models. Converting spectra using GAF and MTF can fully retain the helpful information in the spectrum and better characterize the spectrum through two-dimensional images. RP and RPM transformation methods can interpret the intrinsic relationship between data, provide prior knowledge about similarity and predictability, and facilitate the establishment of predictive models.

To address the issue of increased computational time due to data redundancy in full-spectra analysis, the successive projections algorithm (SPA) [25,26], Boruta and competitive adaptive reweighted sampling (CARS) algorithms were used to extract the feature wavelengths [27,28,29]. SPA is a forward variable selection method that can minimize the collinearity between spectral variables in this work. CARS is a feature variable selection method that combines Monte Carlo sampling with the regression coefficients of the PLS model [30]. It uses the percentage of the absolute value of the regression coefficient as an important indicator to eliminate characteristic wavelength points with redundant information. The Boruta algorithm is a wrapper based on the random forest classification algorithm [31]. The feature extraction method can adaptively handle missing values and noise while reducing the dimensionality of the tube evaluation data, thereby enhancing the robustness of the algorithm.

#### 2.3.3. Data Fusion

According to the fusion structure of multispectral data, the fusion strategies can be divided into three categories: full-spectra fusion, feature-level fusion and decision-level fusion [32,33,34]. Herein, feature-level data fusion is to extract relevant features from individual spectra data sources, respectively, and then combine them into a matrix for processing through modeling methods. Decision-level fusion entails fusing outcomes of classification or regression models from individual techniques to identify the best outcome. Compared with other data fusion strategies, each technique is treated independently in decision-level fusion. Poor performance from one technique does not worsen the overall performance. However, this fusion strategy has not been widely explored. Based on model fusion, decision-level data fusion makes a comprehensive decision on the final results through a voting mechanism, which can be expressed as
(4)$$y_{\mathrm{pred}\;}=k_{1}\times y_{\mathrm{predA}}+k_{2}\times y_{\mathrm{predB}}$$
where $y_{\mathrm{predA}}$ and $y_{\mathrm{predB}}$ are the predictive results of model A and model B. $k_{1}$ and $k_{2}$ are the weight coefficients of $y_{\mathrm{predA}}$ and $y_{\mathrm{predB}}$ determined by the voting mechanism. $y_{\mathrm{pred}}$ is the final comprehensive decision result.

In this work, SERS and FLU spectral data of samples were used to build a quantitative prediction model for potassium sorbate and lead in matsutake based on feature-level and decision-level fusion strategies. The experimental and modeling process of quantitative analysis is shown in Figure 1.

## 3. Results and Discussion

### 3.1. Spectral Curve

Herein, we selected the SERS spectra of four different samples for display. The selected samples are blank extract, extract with potassium sorbate added, extract with lead added and extract with both added. It can be seen from Figure 2a that the Raman peaks at 381 cm^−1^, 903 cm^−1^, 2287 cm^−1^ and 2940 cm^−1^ can be attributed to the extract. The SERS peak at 1049 cm^−1^ belongs to the lead element contained in the extract, and the SERS peak at 883 cm^−1^ and 1651 cm^−1^ belongs to the potassium sorbate. It should be noted that the intensity of the corresponding SERS peaks was independent of the content of the other additive. Therefore, the spectral data at the SERS peaks belonging to the additive were used to build quantitative prediction models. Due to the low intensity and poor spectral discrimination of the SERS peak at 883 cm^−1^, only the SERS peak at 1651 cm^−1^ was used for the quantitative analysis of potassium sorbate. The spectra of the SERS peaks at 1049 cm^−1^ and 1651 cm^−1^ in relation to the corresponding additives are shown in Figure 2b,c. Since there is no interaction in the SERS peaks resulting from lead and the SERS peak belonging to potassium sorbate, the quantitative detection of a binary mixture can be converted to the detection of two one-component additives, which improves the accuracy of quantification. However, when the concentration of potassium sorbate and lead is lower than 0.1 g·kg^−1^ and 0.1 mg·kg^−1^, the characteristic peaks belonging to additives have a high degree of coincidence and are easily overwhelmed by noise, making it difficult for quantitative analysis of low concentrations.

Considering the limitation of the SERS spectra for the detection of low-concentration samples, we employed the FLU spectra. To perform a quantitative analysis of potassium sorbate and lead separately, the samples were tested at different excitation wavelengths according to the test process in the second paragraph. The emission spectra at excitation wavelengths of 262 nm and 358 nm for the corresponding additives are shown in Figure 3a,b. Contrary to the SERS spectra, the intensity of the corresponding FLU emission spectra is negatively correlated with the concentrations of lead and potassium sorbate, respectively, and the emission spectra of low-concentration samples are distinguishable. Similarly, the errand was transformed into quantitative analysis of two one-component substances through sample pre-processing and selection of excitation wavelength.

### 3.2. Modeling and Analysis of the Individual Spectra

The SERS spectral dataset was selected to build quantitative prediction models for potassium sorbate and lead. Due to the strong collinearity of the SERS spectra of the samples, the spectral data of two SERS peaks were selected to establish the prediction model. To avoid generalization errors, the training set and the prediction set were divided into 4:1 by introducing a random function. We convert the detection of binary mixtures into the quantitative analysis of two single substances through the selection of spectra types and wavebands. The complex non-linear characterization errand was transformed into a relatively concise linear characterization errand consequently. Therefore, PLSR was used to establish prediction models. To address the issue of increased computational time and reduced model performance due to data redundancy, we choose the SPA, CARS and Boruta algorithms to further extract characteristic wavelength points within the selected band. Considering the inadequacy of the model in predicting low concentrations where the spectral crossover is severe, DF and 2D CNN models were adopted to further improve the prediction accuracy. Before establishing the 2DCNN model, the extracted one-dimensional spectrum is converted into a two-dimensional spectrum through GAF, MTF, RPM and RP transformation methods to improve the signal-to-noise ratio further and expand the extracted useful information. The performance of CNN is closely related to the appropriate parameter selection. During the modeling process, the Bayes algorithm was introduced to optimize three data-type parameters: mini-batch size, initial learning rate and L2 regularization. The quantitative analysis results of lead element and potassium sorbate using the SERS spectral datasets are given in Table 2.

According to the results in Table 2, it can be found that compared with other prediction models, the prediction results of the CNN-based quantitative prediction model reach higher *R*^2^ and lower *RMSE*, which indicates that the prediction results achieve higher fitting accuracy and more minor errors. Furthermore, the GAF was selected for spectral transformation, which achieved the best performance of the CNN regression model. The model based on the SERS spectra demonstrates relatively stable performance in predicting various kinds of concentrations and can achieve high upper limits of predictability. However, the performance of detecting low-concentration samples was unsatisfactory.

To address the issue of insufficient sensitivity in models based on SERS spectral data, FLU spectral data were used to develop the prediction model. The modeling details were the same as the SERS spectroscopy-based prediction model. Considering the insufficient performance of the MTF, RP and RPM algorithms in transforming 1D spectra into 2D spectrograms to develop SERS spectroscopy-based prediction models, only GAF was used for the establishment of prediction models based on FLU spectra. The modeling results showed that the CNN model based on the CARS feature selection method and the GAF spectral transformation method (CARS-GAF-CNN) was the best quantitative prediction model of potassium sorbate and lead, in which the *R*^2^ were 0.9794 and 0.9743, and *RMSE* were 0.1070 g·kg^−1^ and 0.1117 mg·kg^−1^, respectively. The optimal models for quantifying potassium sorbate and lead elements based on single spectral data are shown in Table 3.

Compared with the potassium sorbate and lead concentration prediction models based on SERS spectral data, the overall accuracy of the prediction model based on FLU spectral data is lower. Because the fluorescence intensity of the sample changes slowly at higher concentrations of potassium sorbate and lead. Moreover, the fluorescence spectra of high-concentration samples are insufficiently stable, making the error in spectral collection more considerable than that of low-concentration samples. These factors reduce the discrimination of fluorescence spectra of high-concentration samples, thereby affecting the predictive performance of the model. However, in the event of concentration lower than 0.1 g·kg^−1^ for the potassium sorbate and 0.1 mg·kg^−1^ for the lead content, respectively, the *MAE* of the model established by the FLU technique is 19.5% and 16.7% lower than that established by the SERS, which demonstrated the advantages of the FLU quantitative model in the low-concentration detection.

The potassium sorbate and lead element content exhibited a linear relationship with the variation of FLU intensity at 444 nm and 318 nm in the corresponding emission spectra, respectively. A standard curve was constructed based on the relationship between the FLU intensity change at each wavelength point on the *Y*-axis and the analyte concentration on the *X*-axis. The standard curves of lead elements and potassium sorbate within the linear concentration range are shown in Figure 4.

A linear relationship was observed between the concentration of potassium sorbate in the range of 0.005 to 1 g·kg^−1^ and the change in FLU intensity at 444 nm in the corresponding emission spectra. The standard curve of potassium sorbate can be expressed as
(5)$$y=44,737.6244\times x_{ps}+301.7523$$
where *y* is the FLU intensity change of the corresponding wavelength point and $x_{ps}$ is the concentration of potassium sorbate at the corresponding wavelength point. In the field of lead concentrations from 0.01 to 0.8 mg·kg^−1^, a linear relationship exists between the concentration of lead and the corresponding change in FLU intensity at 318 nm. The standard curve of lead can be expressed as
(6)$$y=11,726.4395\times x_{l}-220.1624$$
where $x_{l}$ is the concentration of lead at the corresponding wavelength point. The *LOD* in this method can be calculated from the results of the linear fit and the standard deviation of the blank sample measurement. The formula for *LOD* can be expressed as
(7)LOD=3σ/k
where *σ* was the standard deviation of blank sample measurement, and *k* was the slope of the standard calibration curve. According to the formula, the *LOD* in the potassium sorbate and lead element prediction can reach 2.35 mg·kg^−1^ and 9.72 ug·kg^−1^, respectively. Compared with other spectral-based detection methods, the method in this paper can achieve lower detection limits, which is more conducive to the detection of zero-added green agricultural products.

### 3.3. Data Fusion

#### 3.3.1. Modeling and Analysis of Feature-Level Data Fusion

Considering the unsatisfactory performance of the model based on FLU spectroscopy in predicting high concentrations and the lack of precision in predicting low concentrations by the model based on SERS spectroscopy, a fusion approach that combines SERS and FLU spectroscopy was adopted to establish a prediction model. The fusion approach takes advantage of the complementary synergistic advantages of SERS and FLU spectral information to compensate for the shortcomings of a single spectral data source. The full-spectra fusion strategy directly combines multiple low-level features or information during data processing, thereby expanding the adequate information and improving the accuracy of the model. But it will increase the dimension of input information. To avoid the issue of data redundancy, this study has decided to employ feature-level and decision-level data fusion. The relevant features were extracted from SERS and FLU spectra data sources, respectively, and then combined into a matrix for processing through modeling methods. Herein, SPA and CARS feature variable extraction methods were applied to the model establishment due to their superior performance in the prediction models based on SERS and FLU spectral data. Based on the excellent performance in establishing the single-spectra prediction model, CNN was employed to build a feature-level fusion prediction model. The modeling results of feature-level data fusion on FLU and SERS spectra datasets are shown in Table 4.

The results clearly showed CARS-GAF-CNN was the best regression quantitative prediction model of potassium sorbate and lead, in which the *R*^2^ were 0.9903 and 0.9891, and *RMSE* were 0.0848 g·kg^−1^ and 0.0872 mg·kg^−1^, respectively. Due to the fusion of effective information from the two spectra, compared with the model based on a single spectral data, the model based on feature-level data fusion exhibits higher prediction accuracy and shows remarkable stability in predicting various kinds of concentrations. Compared with the prediction model based on the full-spectra fusion strategy, the calculation time of the corresponding model based on the feature-level fusion strategy is significantly reduced. The method has achieved the purpose of efficient and simplified modeling. The *RMSE* of the optimal feature-level fusion models using different feature extraction algorithms were all lower than 0.1, which indicates that feature-level fusion achieved good prediction results.

#### 3.3.2. Modeling and Analysis of Decision-Level Data Fusion

To further improve the predictive accuracy of the models, two spectral models were optimized on the decision level. Decision-level fusion involves the computation of quantitative regression models from each data source and the combination of the results of each model to obtain the final decision. For comparison, two comprehensive evaluation methods, the technique for order preference by similarity to ideal solution (TOPSIS) and the random forest (RF) algorithm, were adopted as voting mechanisms for decision-level fusion [35]. TOPSIS and RF evaluation methods were selected for the establishment of decision-level fusion models due to their fast calculation speed and low susceptibility to outliers. TOPSIS method is a comprehensive decision-making method. The objective assignment of entropy weights is used to calculate the information entropy of the index. The relative change degree of index impact on the whole system determines its weight coefficient. At the same time, the optimal and inferior solutions among the finite solutions can be obtained in the normalized original data matrix. The distances between the evaluated subjects and the two solutions are calculated separately, which can be used as a basis to evaluate the grades of the samples. The RF algorithm can rank the importance by analyzing the magnitude of the contribution made by each feature [36,37]. Variable importance measures (*VIM*) can be expressed by the Gini index (*GI*). The ${\mathrm{GI}}_{q}^{\left(i\right)}$ and ${\mathrm{VIM}}_{jq}^{\left(\mathrm{Gini})(i\right)}$ indicate the Gini index and feature importance of the *i*th tree node *q*. The final normalized importance score for each indicator can be expressed as


(8)
$${\mathrm{VIM}}_{j}^{\left(\mathrm{Gini})(i\right)}=\frac{{\mathrm{VIM}}_{j}^{\left(\mathrm{Gini})(i\right)}}{\sum_{j\prime }^{J}{\mathrm{VIM}}_{j\prime }^{\left(\mathrm{Gini})(i\right)}}$$


Herein, SPA and CARS were applied to the model. Since the PLSR prediction model based on FLU spectra was ineffective in quantitatively predicting high concentrations, a very low weight coefficient was assigned to the predictions of this model in the decision-level data fusion process. The prediction results of the PLSR-based decision-level data fusion model were similar to those of the SERS spectra-based prediction model. Therefore, the predictions of the model based on the PLSR algorithm were not used for advanced fusion, and the best prediction results of the model based on the CNN algorithm were chosen. The optimal results of the prediction model based on SERS and FLU spectral data are recorded as $y_{\mathrm{SERS}}$ and $y_{FLU}$. When establishing the prediction model of potassium sorbate content, the results of decision-level data fusion based on TOPSIS can be expressed as
(9)$$y_{\mathrm{pred}(TOPSIS)}=0.6272\times y_{\mathrm{SERS}}+0.3728\times y_{FLU}$$

The results based on RF can be expressed as
(10)$$y_{\mathrm{pred}(RF)}=0.6683\times y_{SERS}+0.3317\times y_{FLU}$$

When establishing the prediction model of lead, the results of decision-level data fusion based on TOPSIS and RF can be expressed as
(11)$$y_{\mathrm{pred}(TOPSIS)}=0.6766\times y_{\mathrm{SERS}}+0.3234\times y_{FLU}$$
(12)$$y_{\mathrm{pred}(RF)}=0.6683\times y_{SERS}+0.3217\times y_{FLU}$$

Since the prediction results of high-concentration samples have a more significant impact on the overall accuracy of the prediction model, the results of the prediction model based on SERS spectral data that perform better in predicting high concentrations are assigned to higher weights. The modeling results of decision-level data fusion on FLU and SERS spectra datasets are shown in Table 5.

Table 4 clearly showed that the CARS-GAF-CNN model based on the TOPSIS voting mechanism was the best quantitative prediction model of potassium sorbate, in which the *R*^2^ and *RMSE* were 0.9963 and 0.0712 g·kg^−1^. The CARS-GAF-CNN model based on the RF voting mechanism, in which the *R*^2^ and *RMSE* were 0.9934 and 0.0795 mg·kg^−1^, exhibited the best performance in quantitatively analyzing the lead element. Compared with other detection methods of heavy metals in agricultural and sideline products based on spectroscopy and microwave technology, the method in this study improves the detection accuracy [13,38,39]. It can be found that decision-level fusion reduces the impact of weak sensors on the overall model performance by adjusting the weight of results obtained from different sources. It takes advantage of the complementary advantages of quantitative results based on SERS and FLU spectral prediction models to further improve the prediction accuracy of the model. Compared with using the feature-level fusion strategy, the decision-level fusion strategy has little impact on model calculation time and does not violate the original intention of efficient modeling.

To visually compare the models established based on single spectral data with those developed using the data fusion technique, the results of the best models obtained from each approach are presented in Figure 5.

In the quantitative analysis of potassium sorbate and lead, the predictive models achieved optimal results in decision-level data fusion. Compared to the prediction models for potassium sorbate and lead elements based on single-spectra data, the *R*^2^ improves to 0.9963 and 0.9934, and the *RMSE* has decreased by 21.9% and 13.7%, respectively. Overall, the results of the fusion model are better than those of the single spectral model.

## 4. Conclusions

This work focuses on data fusion strategies to improve the prediction accuracy of low-concentration potassium sorbate and lead elements in Tricholoma matsutakes. SERS and FLU spectroscopy were used to quantitatively analyze the potassium sorbate and lead elements simultaneously. By selecting the appropriate waveband and excitation wavelength, we convert the mixed detection of potassium sorbate and lead into the quantitative detection of a single additive to improve the prediction accuracy. Among all the quantitative models, the GAF-CNN model based on decision-level data fusion technology exhibited the best predictive performance, in which the *R*^2^ increased to 0.9963 and 0.9934, and the *RMSE* reduced to 0.0712 g·kg^−1^ and 0.0795 mg·kg^−1^, respectively. It was revealed that decision-level data fusion enormously improved the *R*^2^ and reduced the *RMSE* values. Moreover, the *LOD* of potassium sorbate and lead element can reach 2.35 mg·kg^−1^ and 9.72 ug·kg^−1^, respectively, which can meet the practical applications. The results of this study confirm that building a predictive model based on SERS and FLU spectral data using a decision-level fusion strategy and CNN is an efficient approach for the practical, stable and accurate detection of the quality of Tricholoma matsutakes. However, the method proposed in this study also has limitations. When the analyte concentration is too high, even prediction models based on decision-level data fusion cannot provide accurate quantification due to the instability of the fluorescence spectrum. In addition, the sample pretreatment method used in this study needs to be improved, and the efficiency of detection can be enhanced by simplifying the steps. An online real-time detection system can be developed considering the timeliness of fresh Tricholoma matsutakes samples. In general, the methodology in this study offers rapid and precise detection of the quality of Tricholoma matsutakes based on spectral fusion technology. In the future, this study could be extended to detect and analyze the content of preservatives and heavy metal elements in other precious food ingredients.

## Figures and Tables

**Figure 1 foods-12-04267-f001:**
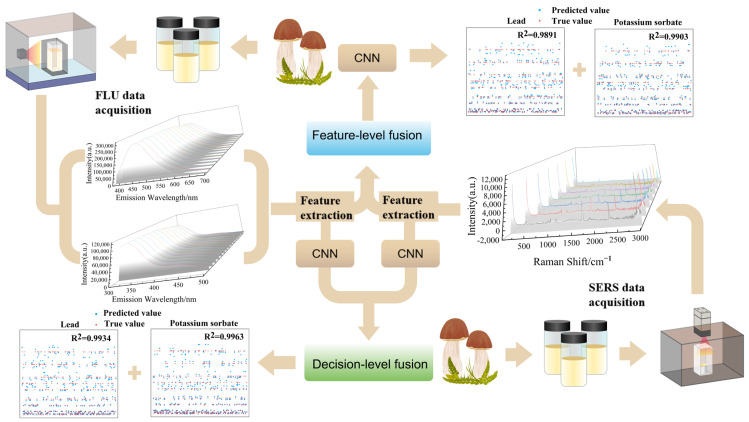
The experimental and modeling process of quantitatively analyzing potassium sorbate and lead in Tricholoma matsutakes.

**Figure 2 foods-12-04267-f002:**
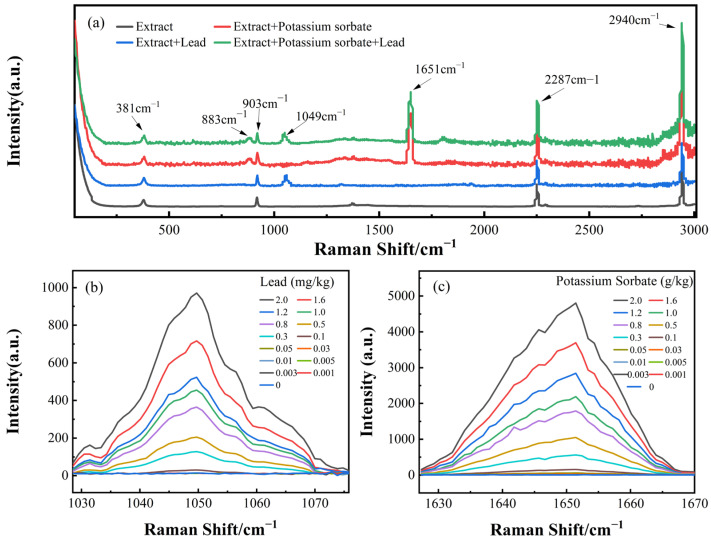
Measured spectra of extracted matsutake and mixture with potassium sorbate and lead. (**a**) The SERS spectra of the exemplary samples and the blank sample. The average spectra of SERS peaks belong to lead (**b**) and potassium sorbate (**c**).

**Figure 3 foods-12-04267-f003:**
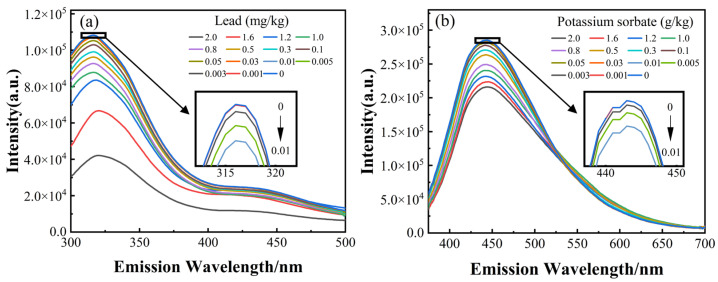
Measured spectra of extracted matsutake and mixture with potassium sorbate and lead. The average FLU emission spectra of samples at excitation wavelengths of 262 nm (**a**) and 358 nm (**b**).

**Figure 4 foods-12-04267-f004:**
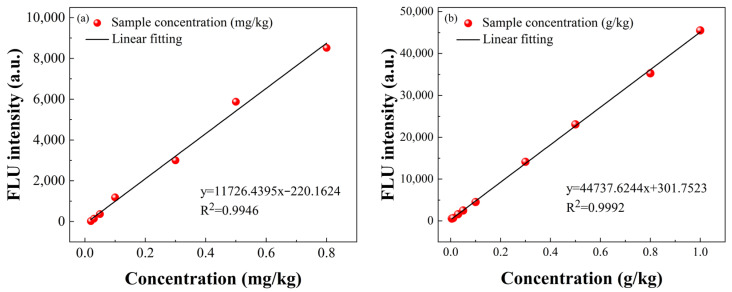
The standard calibration curves for lead (**a**) and potassium sorbate (**b**).

**Figure 5 foods-12-04267-f005:**
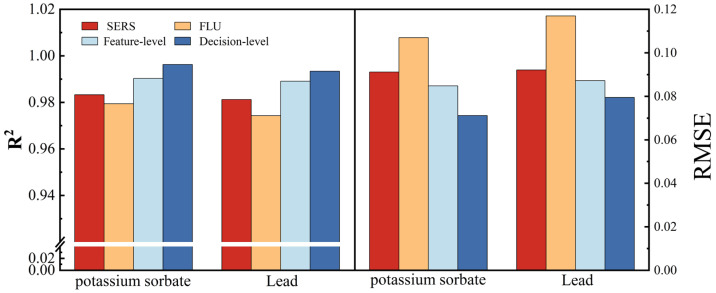
Results of the best models obtained from each approach.

**Table 1 foods-12-04267-t001:** The concentrations of additives in the samples.

Serial Number	Potassium Sorbate (g·kg^−1^)	Lead Element (mg·kg^−1^)
1	0	0
2	0.001	0.001
3	0.003	0.003
4	0.005	0.005
5	0.01	0.01
6	0.03	0.03
7	0.05	0.05
8	0.1	0.1
9	0.3	0.3
10	0.5	0.5
11	0.8	0.8
12	1.0	1.0
13	1.2	1.2
14	1.6	1.6
15	2.0	2.0

**Table 2 foods-12-04267-t002:** Results of lead element and potassium sorbate quantitative prediction models based on SERS spectra datasets.

Methods	Model	Lead Element		Potassium Sorbate	
		*R* ^2^	*RMSE* (mg·kg^−1^)	*R* ^2^	*RMSE* (g·kg^−1^)
none	PLSR	0.9604	0.1227	0.9668	0.1202
SPA		0.9681	0.1172	0.9724	0.1095
Boruta		0.9652	0.1191	0.9688	0.1143
CARS		0.9702	0.1125	0.9725	0.1090
none	DF	0.9677	0.1147	0.9714	0.1109
SPA		0.9714	0.1085	0.9783	0.1026
Boruta		0.9685	0.1097	0.9735	0.1078
CARS		0.9725	0.1066	0.9803	0.0997
SPA-GAF	CNN	0.9801	0.0894	0.9833	0.0841
Boruta-GAF		0.9782	0.0972	0.9781	0.0931
CARS-GAF		0.9812	0.0875	0.9829	0.0852
SPA-MTF		0.9741	0.1012	0.9779	0.0967
Boruta-MTF		0.9688	0.1097	0.9751	0.1002
CARS-MTF		0.9748	0.0962	0.9775	0.0972
SPA-RP		0.9785	0.0923	0.9812	0.0895
Boruta-RP		0.9698	0.1067	0.9766	0.1021
CARS-RP		0.9792	0.0901	0.9810	0.0899
SPA-RPB		0.9765	0.0992	0.9804	0.0907
Boruta-RPB		0.9724	0.1075	0.9789	0.0931
CARS-RPB		0.9766	0.0990	0.9799	0.0918

**Table 3 foods-12-04267-t003:** The best prediction results based on each individual spectra.

Analyte	Spectra	Models	*R* ^2^	*RMSE*
Lead element	SERS	CARS-GAF-CNN	0.9812	0.0875
Potassium sorbate		SPA-GAF-CNN	0.9833	0.0841
Lead element	FLU	CARS-GAF-CNN	0.9743	0.1117
Potassium sorbate		CARS-GAF-CNN	0.9794	0.1070

The *RMSE* of the quantitative model for potassium sorbate and the quantitative model for the lead element is expressed in g·kg^−1^ and mg·kg^−1^, respectively.

**Table 4 foods-12-04267-t004:** Results of quantitative prediction models based on feature-level fusion.

Analyte	Potassium Sorbate		Lead Element	
Methods	SPA-CNN	CARS-CNN	SPA-CNN	CARS-CNN
*R* ^2^	0.9881	0.9903	0.9852	0.9891
*RMSE*	0.0902 g·kg^−1^	0.0848 g·kg^−1^	0.0908 mg·kg^−1^	0.0872 mg·kg^−1^

**Table 5 foods-12-04267-t005:** Results of quantitative prediction models based on decision-level fusion.

Analyte	Potassium Sorbate		Lead Element	
Methods	TOPSIS-CNN	RF-CNN	TOPSIS-CNN	RF-CNN
*R* ^2^	0.9963	0.9952	0.9932	0.9934
*RMSE*	0.0712 g·kg^−1^	0.0741 g·kg^−1^	0.0803 mg·kg^−1^	0.0795 mg·kg^−1^

## Data Availability

The data presented in this study are available on request from the corresponding author. The data are not publicly available due to ongoing funding projects which not provide public data sharing before the end of the project.

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
