# Peer review of "Simultaneous Quantitative Determination of Low-Concentration Preservatives and Heavy Metals in Tricholoma Matsutakes Based on SERS and FLU Spectral Data Fusion"

_foods, 2023, doi:10.3390/foods12234267_

Round 1
Reviewer 1 Report
Comments and Suggestions for Authors
The abstract could benefit from improved clarity and organization. It currently presents information in a somewhat disjointed manner. Consider rephrasing and restructuring sentences to make the content flow more smoothly and logically. The introduction should provide more context regarding the significance of the research. Why is it essential to detect potassium sorbate and lead in Tricholoma matsutakes? What are the potential health implications? Providing a clear background will help readers understand the importance of the study. The section on data fusion is somewhat technical and may require simplification or additional explanations for readers who are not experts in the field. Clarify how data fusion works and its relevance to the study. The methods section is quite technical, which might make it challenging for non-experts to follow. Provide more detailed explanations or references for the spectroscopy techniques used, such as SERS and FLU. Additionally, explain the significance of Gramian angular field (GAF), Markov transition field (MTF), etc., in simpler terms. When discussing feature extraction methods like SPA, Boruta, and CARS, it might be helpful to provide a brief overview of what these techniques do and why they are relevant. When discussing data fusion, clarify why feature-level data fusion and decision-level data fusion are essential for this study. Provide examples or explanations to illustrate the concept. Explain in more detail how the correlation coefficient is calculated using the TOPSIS and RF methods. Clarify why these methods were chosen. Consider breaking down complex concepts into simpler language where possible. Ensure that the text is accessible to a broader audience, not just experts in the field. The results section begins with a description of the data sources (SERS and FLU spectra) and then discusses various modeling approaches. While the content is relevant, the section lacks a clear structure. Consider reorganizing it to provide a more logical flow, such as first describing the data sources and then presenting the modeling approaches. Ensure that the names of the models are clearly explained and defined upon first mention. For example, define "CARS-GAF-CNN" and "TOPSIS-GAF-CNN" when they are first introduced in the text to help readers understand the model components. When discussing the data fusion process, provide a brief overview of how the decision-level data fusion is conducted, including the weight assignment or combination method used. This will help readers understand the basis for combining results from different sources. Include clear tables or figures summarizing the results of each model to facilitate comparison. It can be beneficial to have a clear visual representation of the performance metrics, such as R2 and RMSE, for each model. For each model, provide a brief interpretation of what the R2 and RMSE values mean in the context of the problem. Explain the practical significance of achieving certain levels of accuracy. Discuss the limitations of the models and potential sources of error. For instance, you mention that the FLU-based models were less effective for high concentrations. Explain the reasons behind this limitation and potential steps for improvement. Discuss the practical applicability of the models. How could these models be used in real-world scenarios for detecting potassium sorbate and lead in Tricholoma matsutakes? What are the implications of achieving low LOD values? Consider adding a subsection on future work. What are the potential next steps or improvements in this research? Are there additional factors or elements that could be incorporated into the models? Ensure that you cite relevant references to support your claims and provide context for the methods and models used. If there are specific algorithms or techniques you've employed, referencing the sources or studies where they were developed or explained can be helpful.
By addressing these points, you can improve the clarity and readability of your "Results and Discussion" section and provide a more comprehensive understanding of your research to your readers.
Comments on the Quality of English LanguageModerate editing of English language required
Author Response
For research article
|
Response to Reviewer 1 Comments
|
||
|
1. Summary |
|
|
|
We deeply appreciate the reviewer’s comment on our work. Below we address the comments and queries raised by the referees (Comments from the referees are italicized). We hope that the revised manuscript has taken full account of the points raised by the reviewers.
|
||
|
2. Questions for General Evaluation |
Reviewer’s Evaluation |
Response and Revisions |
|
Does the introduction provide sufficient background and include all relevant references? |
Can be improved |
|
|
Are all the cited references relevant to the research? |
Can be improved |
|
|
Is the research design appropriate? |
Must be improved |
|
|
Are the methods adequately described? |
Must be improved |
|
|
Are the results clearly presented? |
Must be improved |
|
|
Are the conclusions supported by the results? |
Must be improved |
|
|
3. Point-by-point response to Comments and Suggestions for Authors |
||
|
Comments 1: The abstract could benefit from improved clarity and organization. It currently presents information in a somewhat disjointed manner. Consider rephrasing and restructuring sentences to make the content flow more smoothly and logically.
|
||
|
Response 1: Thank you for pointing this out. We are grateful for your comment. We modify the abstract to make it more coherent. Abstract: As an ingredient of great economic value, Tricholoma matsutake has received widespread attention. However, heavy metal residues and preservatives in it will affect the quality of Tricholoma matsutake and endanger the health of consumers. Here, we present a method for the simultaneous detection of low concentrations of potassium sorbate and lead in Tricholoma matsutakes based on surface-enhanced Raman spectroscopy (SERS) and fluorescence (FLU) spectroscopy to test the safety of consumption. Data fusion strategies combined with multiple machine learning methods, including partial least squares regression (PLSR), deep forest (DF) and convolutional neural networks (CNN), are used for model training. The results show that combined with reasonable band selection, the CNN prediction model based on decision-level fusion achieves the best performance, the correlation coefficients (R2) were increased to 0.9963 and 0.9934, and the root mean square errors (RMSE) were reduced to 0.0712 g·kg-1 and 0.0795 mg·kg-1, respectively. The method proposed in this paper accurately predicts preservatives and heavy metals remaining in Tricholoma matsutake and provides a reference for other food safety testing. This change can be found in the revised manuscript – page 1, paragraph 0, and lines 9-21.
|
||
|
Comments 2: The introduction should provide more context regarding the significance of the research. Why is it essential to detect potassium sorbate and lead in Tricholoma matsutakes? What are the potential health implications? Providing a clear background will help readers understand the importance of the study.
|
||
|
Response 2: We deeply appreciate your comment. The detection of preservatives and heavy metal content plays a vital role in the quality control of Tricholoma matsutakes and the guarantee of food safety. We cover more dangers of excessive preservatives and heavy metal intake. The description “The accumulation of preservatives and heavy metals in the body may lead to an acid-base balance imbalance in the human body, causing symptoms such as dizziness and diarrhoea. In severe cases, it may cause chronic poisoning and increase the risk of cancer. To maintain good health, people should pay attention to controlling their daily intake of preservatives and heavy metals. Therefore, the detection of preservatives and heavy metal content plays a vital role in the quality control of Tricholoma matsutakes and the guarantee of food safety.” is added to lines 36-42 of page 1.
|
||
|
Comments 3: The section on data fusion is somewhat technical and may require simplification or additional explanations for readers who are not experts in the field. Clarify how data fusion works and its relevance to the study.
|
||
|
Response 3: We are grateful for your valuable suggestions. In the revised manuscript, we have added a description of how data fusion works and its relevance to the study. The description “SERS-based methods require considerable efforts to develop corresponding substrates for analytes to enhance the Raman signal and improve the accuracy of prediction. Compared with the SERS-based method, the FLU-based method can achieve higher sensitivity and resolution during measurement. Affected by factors such as scattering, self-absorption, and temperature, the instability of fluorescence methods at high concentrations will limit the accuracy of the prediction model. With the increased emphasis on food safety and the strict restrictions on the content of food additives, achieving simultaneous detection of multiple mixed substances while ensuring detection accuracy has become a hot spot in the current field of food testing. Therefore, the spectral data fusion technology is used in this work to make up for the shortcomings of single-spectral methods to establish a simultaneous detection model for preservatives and heavy metals.” is added to lines 64-75 of page 2 to explain the relevance of data fusion to this study. We supplement the introduction of fusion methods before using fusion techniques. The supplement “The relevant features were extracted from SERS and FLU spectra data sources, respectively, and then combined into a matrix for processing through modeling methods.” is added to lines 383-385 of page 11.
|
||
|
Comments 4: The methods section is quite technical, which might make it challenging for non-experts to follow. Provide more detailed explanations or references for the spectroscopy techniques used, such as SERS and FLU.
|
||
|
Response 4: We sincerely appreciate your reminder. We are very sorry for the unclear introduction to spectroscopy technology. We have added an introduction to the theory of SERS and FLU spectroscopy, hoping that non-expert readers can have a better understanding of SERS and FLU spectroscopy. The introduction “Raman spectroscopy is a powerful label-free technique to identify molecules by measuring the vibrational and rotational character of their chemical bonds. SERS exploits the phenomenon of enhanced Raman scattering on the surface of plasmonic nanoparticles or nanostructures. Fluorescence is a type of radiation transition, which is the radiation released by a substance from an excited state to a low-energy state with the same multiplicity. When a molecule in the ground state absorbs energy and jumps, the molecule becomes unstable and jumps back to the ground state. Photons are emitted during the transition back to the ground state, which produces fluorescence.” is added to lines 141-144 and 153-157 of page 4.
|
||
|
Comments 5: Additionally, explain the significance of Gramian angular field (GAF), Markov transition field (MTF), etc., in simpler terms.
|
||
|
Response 5: Thank you for pointing this out. We are grateful for your reminders. The extracted one-dimensional spectrum is converted into a two-dimensional spectrum through GAF, MTF, RPM and RP transformation methods to further improve the signal-to-noise ratio and expand the extracted useful information. The supplement “Converting spectra using GAF and MTF can fully retain the helpful information in the spectrum and better characterize the spectrum through two-dimensional images. RP and RPM transformation methods can interpret the intrinsic relationship between data, provide prior knowledge of similarity and predictability, and facilitate the establishment of predictive models.” is added to lines 197-201 of page 5 to explain the reasons for choosing these conversion methods.
|
||
|
Comments 6: When discussing feature extraction methods like SPA, Boruta, and CARS, it might be helpful to provide a brief overview of what these techniques do and why they are relevant.
|
||
|
Response 6: We appreciate your in-depth guidance on the article We choose the feature extraction algorithms to further extract characteristic wavelength points within the previously selected band to eliminate redundant data and reduce the calculation time of the models. We have added an introduction to the role of feature extraction methods in spectral data processing in this study. The overview “SPA is a forward variable selection method that can minimize the collinearity between spectral variables in this work. CARS is a feature variable selection method that combines Monte Carlo sampling with the regression coefficients of the PLS model. It uses the percentage of the absolute value of the regression coefficient as an important indicator to eliminate characteristic wavelength points with redundant information. The Boruta algorithm is a wrapper based on the random forest classification algorithm. The feature extraction method can adaptively handle missing values and noise while reducing the dimensionality of the tube evaluation data, thereby enhancing the robustness of the algorithm.” is added to lines 205-213 of page5.
|
||
|
Comments 7: When discussing data fusion, clarify why feature-level data fusion and decision-level data fusion are essential for this study. Provide examples or explanations to illustrate the concept.
|
||
|
Response 7: We deeply appreciate your comment. We introduce the role of data fusion techniques in this study and provide more examples to demonstrate the feasibility of spectral fusion strategy in quantitative analysis. The example “Zhao, et al. used near-infrared (NIR) and Laser-induced breakdown spectroscopy (LIBS) to quantitatively analyze the heavy metals in lily. The introduction of near-infrared spectroscopy makes up for the inability of LIBS to accurately quantify complex matrix samples. Compared with the full-spectrum model, the model based on feature-level fusion achieves better performance in quantifying Zn, Cu and Pb, with R2 of 0.9858, 0.9811 and 0.9460, and RMSE of 4.3047 mg·kg-1, 4.9592 mg·kg-1 and 0.9460 mg·kg-1.” is added to line 80-86 of page 2. We mentioned “The full-spectra fusion strategy directly combines multiple low-level features or information during data processing, thereby expanding the adequate information and improving the accuracy of the model. But it will increase the dimension of input information. To avoid the issue of data redundancy, this study has decided to employ feature-level and decision-level data fusion. The relevant features were extracted from SERS and FLU spectra data sources, respectively, and then combined into a matrix for processing through modeling methods.” in lines 379-385 of page 10-11 to clarify why feature-level data fusion and decision-level data fusion are essential for this study.
|
||
|
Comments 8: Explain in more detail how the correlation coefficient is calculated using the TOPSIS and RF methods. Clarify why these methods were chosen. Consider breaking down complex concepts into simpler language where possible. Ensure that the text is accessible to a broader audience, not just experts in the field.
|
||
|
Response 8: Thank you for pointing this out. We are very sorry for the unclear description of the correlation coefficient calculation method. We explain in more detail how to calculate the correlation coefficient using the TOPSIS and RF methods and the reasons for the choice. The explanation “For comparison, two comprehensive evaluation methods, the technique for order preference by similarity to ideal solution (TOPSIS) and the random forest (RF) algorithm, were adopted as voting mechanisms for decision-level fusion. TOPSIS and RF evaluation methods were selected for the establishment of decision-level fusion models due to their fast calculation speed and low susceptibility to outliers. TOPSIS method is a comprehensive decision-making method. The objective assignment of entropy weights is used to calculate the information entropy of the index. The relative change degree of index impact on the whole system determines its weight coefficient. At the same time, the optimal and inferior solutions among the finite solutions can be obtained in the normalized original data matrix. The distances between the evaluated subjects and the two solutions are calculated separately, which can be used as a basis to evaluate the grades of the samples. The RF algorithm can rank the importance by analyzing the magnitude of the contribution made by each feature. Variable importance measures (VIM) can be expressed by the Gini index (GI). The and indicate the Gini index and feature importance of the ith tree node q. The final normalized importance score for each indicator can be expressed as ” is added to lines 411-425 of page 11. (formula cannot be entered here)
|
||
|
Comments 9: The results section begins with a description of the data sources (SERS and FLU spectra) and then discusses various modeling approaches. While the content is relevant, the section lacks a clear structure. Consider reorganizing it to provide a more logical flow, such as first describing the data sources and then presenting the modeling approaches.
|
||
|
Response 9: Thank you for pointing this out. We are grateful for your advice. We improve the structure of the Discussion section. We first analyzed the SERS and FLU spectral data sources and then gradually presented the modeling approaches based on single spectral data and the modeling approaches based on data fusion strategy. We added connectives between each paragraph to make the structure clearer. This change can be found in the revised manuscript – page 6-13 and lines 237-477.
|
||
|
Comments 10: Ensure that the names of the models are clearly explained and defined upon first mention. For example, define "CARS-GAF-CNN" and "TOPSIS-GAF-CNN" when they are first introduced in the text to help readers understand the model components.
|
||
|
Response 10: We are grateful for your reminders. We are very sorry for the wrong representation. We define "CARS-GAF-CNN" when it is first introduced in the text and modify the expression of "TOPSIS-GAF-CNN". This change can be found in the revised manuscript – page 9, paragraph 2, and lines 320-321. page 12, paragraph 2, and lines 449-4453.
|
||
|
Comments 11: When discussing the data fusion process, provide a brief overview of how the decision-level data fusion is conducted, including the weight assignment or combination method used. This will help readers understand the basis for combining results from different sources. Include clear tables or figures summarizing the results of each model to facilitate comparison.
|
||
|
Response 11: We are grateful for your suggestion. We supplement an overview of how the decision-level data fusion is conducted, including the weight assignment or combination method used. The overview “For comparison, two comprehensive evaluation methods, the technique for order preference by similarity to ideal solution (TOPSIS) and the random forest (RF) algorithm, were adopted as voting mechanisms for decision-level fusion. TOPSIS and RF evaluation methods were selected for the establishment of decision-level fusion models due to their fast calculation speed and less susceptibility to outliers. TOPSIS method is a comprehensive decision-making method. The objective assignment of entropy weights is used to calculate the information entropy of the index. The relative change degree of index impact on the whole system determines its weight coefficient. At the same time, the optimal and inferior solutions among the finite solutions can be obtained in the normalized original data matrix. The distances between the evaluated subjects and the two solutions are calculated separately, which can be used as a basis to evaluate the grades of the samples. The RF algorithm can rank the importance by analyzing the magnitude of the contribution made by each feature. Variable importance measures (VIM) can be expressed by the Gini index (GI). The and indicate the Gini index and feature importance of the ith tree node q.” is added to lines 408-423 of page 11. (formula cannot be entered here) Because the best prediction results of the single spectral model were selected when performing decision-level data fusion, there are a total of four results when quantifying potassium sorbate and lead under both voting mechanisms. All results of models based on the decision-level fusion have been shown in Table 5.
|
||
|
Comments 12: It can be beneficial to have a clear visual representation of the performance metrics, such as R2 and RMSE, for each model. For each model, provide a brief interpretation of what the R2 and RMSE values mean in the context of the problem. Explain the practical significance of achieving certain levels of accuracy.
|
||
|
Response 12: Thank you for pointing this out. We are grateful for your reminders. We have added tables to ensure that the model results mentioned in the article are displayed in the form of visual charts. As important indicators to evaluate the prediction accuracy of the model, in this article, R2 is used to evaluate the degree of fit of the model to the data, and RMSE is used to represent the direct error between the predicted values of potassium sorbate and lead content and the actual values. We explain the meaning of R2 and RMSE when analyzing the forecast results, indicating that in actual forecasts, the higher the R2 and the lower the RMSE, the higher the accuracy of the forecast results. This change can be found in the revised manuscript – page 9, paragraph 1, and lines 305-308.
|
||
|
Comments 13: Discuss the limitations of the models and potential sources of error. For instance, you mention that the FLU-based models were less effective for high concentrations. Explain the reasons behind this limitation and potential steps for improvement.
|
||
|
Response 13: Thank you for pointing this out. We are grateful for your reminders. We have added discussion of the limitations of the FLU-based models and potential sources of error. The discussion “Because the fluorescence intensity of the sample changes slowly at higher concentrations of potassium sorbate and lead. Moreover, the fluorescence spectra of high-concentration samples are insufficiently stable, making the error in spectral collection larger than that of low-concentration samples. These factors reduce the discrimination of fluorescence spectra of high-concentration samples, thereby affecting the predictive performance of the model.” is added to lines 331-336 of page 9. Limited by the experimental instruments and the characteristics of the fluorescence spectrum, the model cannot be improved from a single spectral perspective. We make up for the lack of FLU spectral information by introducing SERS spectral information for data fusion.
|
||
|
Comments 14: Discuss the practical applicability of the models. How could these models be used in real-world scenarios for detecting potassium sorbate and lead in Tricholoma matsutakes? What are the implications of achieving low LOD values?
|
||
|
Response 14: Thank you for pointing this out. We are grateful for your reminders. In practical applications, the spectral data of matsutake samples are directly brought into the established model to quantify the potassium sorbate and lead contents. Achieving low LOD values aims at completing the detection of trace amounts of potassium sorbate and lead in Tricholoma matsutakes. Trace residue detection is meaningful for testing the quality of so-called zero-added green agricultural products sold by merchants. The method of detecting trace residues can also provide a reference for the quantification of other substances harmful to the human body.
|
||
|
Comments 15: Consider adding a subsection on future work. What are the potential next steps or improvements in this research? Are there additional factors or elements that could be incorporated into the models?
|
||
|
Response 15: We deeply appreciate your suggestions on experimental prospects. The potential next steps and improvements in this research are added to the conclusion. The outlook for the research “When the analyte concentration is too high, even prediction models based on decision-level data fusion cannot provide accurate quantification due to the instability of the fluorescence spectrum. In addition, the sample pretreatment method used in this study needs to be improved, and the efficiency of detection can be enhanced by simplifying the steps. An online real-time detection system can be developed considering the timeliness of fresh Tricholoma matsutakes samples.” is added to lines 492-498 of page 13.
|
||
|
Comments 16: Ensure that you cite relevant references to support your claims and provide context for the methods and models used. If there are specific algorithms or techniques you've employed, referencing the sources or studies where they were developed or explained can be helpful.
|
||
|
Response 16: Thank you for pointing this out. We appreciate your reminder of the lack of elaboration of the background and rationale for the algorithms used in this research. We cite additional references to support our chosen methods. We cite other references to ensure that the algorithms and models used in the research in this article are well-founded.
|
||
|
4. Response to Comments on the Quality of English Language |
||
|
Point 1: Moderate editing of English language required |
||
|
Response 1: We are grateful for your comments and suggestions on the Quality of English Language. We have carefully checked and improved the English writing in the revised manuscript.
|
||
|
5. Additional clarifications |
||
|
We greatly appreciate the editor and the reviewers for their efforts on our manuscripts. We are very grateful to the reviewers for their valuable comments on our manuscript. Now we have revised the manuscript and addressed all the issues raised by the reviewer. |
||

Reviewer 2 Report
Comments and Suggestions for Authors
This study introduces a method utilizing surface-enhanced Raman and fluorescence spectroscopy for concurrently identifying trace amounts of potassium sorbate as a preservative and lead as a heavy metal in Tricholoma matsutakes mushroom. It is a well-written manuscript, but the following points should be corrected.
• Line 36: Illegal additives, are you referring to potassium sorbate? I don’t think it is an illegal additive.
• Line 107: “our country” should be more straightforward. I see the authors are from China, but this standard should be stated better.
• Line 111: Another word should be used instead of “kind”. Because you are applying different applications, instead of saying “different kinds of samples,” something else can be used.
• Lines 96-112: How did you homogenize the samples? Did you use any liquid addition, use a blender, etc.? This section whole should be explained in detail.
• Did you validate your models by using an external validation set or how did you validate your findings?
• Are there any outliers that you excluded from the models?
• Are your findings comparable with the literature? There is not much literature comparison in the discussion section of the manuscript.
Comments on the Quality of English LanguageThe manuscript was easy to follow and understand
Author Response
For research article
|
Response to Reviewer 2 Comments
|
||
|
1. Summary |
|
|
|
We deeply appreciate the reviewer’s comment on our work. Below we address the comments and queries raised by the referees (Comments from the referees are italicized). We hope that the revised manuscript has taken full account of the points raised by the reviewers.
|
||
|
2. Questions for General Evaluation |
Reviewer’s Evaluation |
Response and Revisions |
|
Does the introduction provide sufficient background and include all relevant references? |
Can be improved |
|
|
Are all the cited references relevant to the research? |
Can be improved |
|
|
Is the research design appropriate? |
Can be improved |
|
|
Are the methods adequately described? |
Can be improved |
|
|
Are the results clearly presented? |
Can be improved |
|
|
Are the conclusions supported by the results? |
Can be improved |
|
|
3. Point-by-point response to Comments and Suggestions for Authors |
||
|
Comments 1: Line 36: Illegal additives, are you referring to potassium sorbate? I don’t think it is an illegal additive.
|
||
|
Response 1: Thank you for pointing this out. We agree with this comment. We are very sorry for the wrong expression. Potassium sorbate itself is not an illegal additive, but excessive use is illegal in China. Therefore, we changed “illegal additives” to “illegally used additives”. This change can be found in the revised manuscript – page 2, paragraph 1, and line 44.
|
||
|
Comments 2: Line 107: “our country” should be more straightforward. I see the authors are from China, but this standard should be stated better.
|
||
|
Response 2: We deeply appreciate your reminder. We apologize for the unclear description. We should write the country to which the regulations belong. Therefore, we changed “in our country” to “in China”. This change can be found in the revised manuscript – page 3, paragraph 2, and line 132.
|
||
|
Comments 3: Line 111: Another word should be used instead of “kind”. Because you are applying different applications, instead of saying “different kinds of samples,” something else can be used.
|
||
|
Response 3: Thank you for pointing this out. We appreciate your suggestion. We are very sorry for the unclear expression. “different kinds of samples” means that the prepared samples have 225 different concentrations. Therefore, we changed “By combining different concentrations of potassium sorbate and lead, 225 different kinds of samples were prepared in the experiment” to “By combining different concentrations of potassium sorbate and lead, samples were prepared at 225 different concentrations in the experiment”. This change can be found in the revised manuscript– page 3, paragraph 2, and line 136.
|
||
|
Comments 4: Lines 96-112: How did you homogenize the samples? Did you use any liquid addition, use a blender, etc.? This section whole should be explained in detail.
|
||
|
Response 4: We are grateful for your advice. We use a ceramic knife to homogenize the matsutake into small particles. We have added acetonitrile to the samples during extraction. We further describe the sample preparation procedure. The description “Collected Tricholoma matsutakes were cleaned with ultrapure water and homogenized into small particles by a ceramic knife. The potassium sorbate aqueous solution and lead standard solution were added proportionally to the cleaned and homogenized samples to simulate contamination of preservatives and heavy metal elements. We added acetonitrile and extraction salt to the homogenized samples, and took the supernatant as the Tricholoma matsutake extract, and vortexed the extract in the purification tube to eliminate fluorescence interference.” is added to lines 119-125 of page 3.
|
||
|
Comments 5: Did you validate your models by using an external validation set or how did you validate your findings?
|
||
|
Response 5: Thank you for pointing this out. We deeply appreciate your comment. We divide the data set at a ratio of 1:4, and 20% of the data is divided into the test set. In addition, we separately collected spectral data from 10 new samples to verify the model.
|
||
|
Comments 6: Are there any outliers that you excluded from the models?
|
||
|
Response 6: We appreciate your careful review of our paper and your questions. We only excluded outliers when acquiring spectral data, and all prediction results of the models were retained.
|
||
|
Comments 7: Are your findings comparable with the literature? There is not much literature comparison in the discussion section of the manuscript.
|
||
|
Response7: We are grateful for your suggestion. Since there are few applications of spectroscopic technology in the detection of additives in edible fungi, we cited references to compare the research results with the detection results of other agricultural and sideline products. Compared with the prediction results of other studies, the prediction error based on the method in this study is more minor. This change can be found in the revised manuscript – page 12, paragraph 2, and line 456. Reference [13,38,39].
|
||
|
4. Response to Comments on the Quality of English Language |
||
|
Point 1: |
||
|
Response 1: We are grateful for your comments and suggestions on the Quality of English Language. We have carefully checked and improved the English writing in the revised manuscript.
|
||
|
5. Additional clarifications |
||
|
We greatly appreciate the editor and the reviewers for their efforts on our manuscripts. We are very grateful to the reviewers for their valuable comments on our manuscript. Now we have revised the manuscript and addressed all the issues raised by the reviewer. |
||